# Impact of T Cell Exhaustion and Stroma Senescence on Tumor Cell Biology and Clinical Outcome of Head and Neck Squamous Cell Carcinomas

**DOI:** 10.3390/ijms252413490

**Published:** 2024-12-17

**Authors:** Lukas A. Brust, Meike Vorschel, Sandrina Körner, Moritz Knebel, Jan Philipp Kühn, Silke Wemmert, Sigrun Smola, Mathias Wagner, Bernhard Schick, Maximilian Linxweiler

**Affiliations:** 1Department of Otorhinolaryngology, Head and Neck Surgery, Saarland University, 66421 Homburg, Germany; vorschelm@gmail.com (M.V.); sandrina.koerner@uks.eu (S.K.); moritz.knebel@uks.eu (M.K.); jan.kuehn@uks.eu (J.P.K.); silke.wemmert@uks.eu (S.W.); bernhard.schick@icloud.com (B.S.); maximilian.linxweiler@uks.eu (M.L.); 2Institute of Virology, Saarland University, 66421 Homburg, Germany; sigrun.smola@uks.eu; 3Department of General and Surgical Pathology, Saarland University, 66421 Homburg, Germany; mathias.wagner@uks.eu

**Keywords:** HNSCC, biomarker, TME, HPV, PD-1, LAG-3, TIM-3, IL-8, MMP-3

## Abstract

Head and neck squamous cell carcinomas (HNSCC) have an overall poor prognosis, especially in locally advanced and metastatic stages. In most cases, multimodal therapeutic approaches are required and show only limited cure rates with a high risk of tumor recurrence. Anti-PD-1 antibody treatment was recently approved for recurrent and metastatic cases but to date, response rates remain lower than 25%. Therefore, the investigation of the immunological tumor microenvironment and the identification of novel immunotherapeutic targets in HNSCC is of paramount importance. In our study, we used tissue samples of n = 116 HNSCC patients for the immunohistochemical detection of the intratumoral and peritumoral expression of T cell exhaustion markers (PD-1, LAG-3, TIM-3) on tumor infiltration leukocytes (TIL), as well as the expression level of stromal senescence markers (IL-8, MMP-3) on tumor-associated fibroblasts. The clinical parameter of the vitamin D serum status as well as the histopathological HPV infection status of the tumor was correlated with the expression rates of the biomarkers and the overall patient survival. An increased peritumoral and intratumoral expression of the biomarkers PD-1 and TIM-3 significantly correlated with improved overall patient survival. A high peritumoral expression of LAG-3 correlated with better overall survival. A positive HPV tumor status correlated with a significantly elevated expression of PD-1 and TIM-3. Biomarkers of stromal senescence showed no influence on the patient outcome. However, the vitamin D serum status showed no influence on patient outcomes or biomarker expressions. Our study identified PD-1, LAG-3, and TIM-3 as promising targets of a therapeutic strategy targeting the tumor microenvironment in HNSCC, particularly among HPV-positive patients, where a higher expression of these checkpoints correlated with an improved overall survival. These findings support the potential of antibodies targeting these immune checkpoints to enhance treatment efficacy, especially in the context of bispecific targeting.

## 1. Introduction

Head and neck squamous cell carcinomas (HNSCC) are the sixth-most common cancer type globally, with 895,000 new cases and 457,000 deaths in 2022 [1]. Major risk factors include chronic nicotine and alcohol consumption, with a growing contribution from high-risk HPV infection, particularly in oropharyngeal cancers [2]. Locally advanced and metastatic HNSCC stages maintain a poor prognosis, with five-year survival rates of approximately 60% [3].

The majority of HNSCC patients require multimodal treatment, including surgery, radiation, and chemotherapy. However, tumor recurrence is common [4] with highly limited therapeutic approaches for this challenging clinical setting. Recently, two anti-PD-1 immune checkpoint inhibitors (ICIs), pembrolizumab and nivolumab, have been approved for the treatment of recurrent and metastatic HNSCC [5] for first-line [6] and second-line treatment either as a monotherapy or in combination with platinum-based chemotherapy [7,8]. However, response rates remain at a low level, around 25%, and resistance to immune-checkpoint inhibition is frequently observed over time. The limited clinical success of the currently approved immunotherapeutic strategies underscore the need for further research in head and neck cancer immuno-oncology, particularly as HNSCC is among the most immune-infiltrated human cancers [9,10]. In the context of possible adjuvants, vitamin D has been shown in preclinical and retrospective clinical studies to be a positive prognostic marker, as well as a possible agent for increasing response rates due to its immunomodulatory properties [11,12]. Furthermore, the molecular processes of T cell exhaustion and stroma senescence as potential mechanisms of evading immune surveillance and an antitumoral immune response came into the spotlight over the past years with, however, only limited data in head and neck cancer so far [13,14].

T cell exhaustion represents a state of T cell dysfunction that arises during chronic infections and cancer development. It is defined by poor T cell effector function, a high expression of inhibitory checkpoint receptors including CTLA-4, PD-1, TIM-3, BTLA, VISTA, and LAG-3 [15], and an altered transcriptional program. LAG-3, a key checkpoint receptor alongside PD-1 and CTLA-4, promotes tumor growth by inhibiting the immune response at high expression levels [15]. Similarly, TIM-3, which is highly expressed in tumor-infiltrating lymphocytes, suppresses anti-tumor immunity through its interaction with Galectin 9 in various cancers [16]. Overall, T cell exhaustion was shown to be associated with the ineffective immunological control of chronic infections and several cancer types [17,18] including melanoma [19], chronic myeloid leukemia [20], ovarian cancer [21], and non-small cell lung cancer [22].

In addition to tumor-infiltrating leucocytes, peritumoral stromal tissue represents the major component of the tumor microenvironment (TME). Peritumoral stroma primarily consists of cancer-associated fibroblasts (CAFs) and an extracellular matrix (ECM) and was shown to have a relevant role in cancer progression through cell–cell and cell–matrix interactions in different cancer types [23]. Cancer development and progression as well as tumor treatment can induce stromal changes and lead to the accumulation of senescent stromal cells that are characterized by the so-called senescence-associated secretory phenotype (SASP). Those senescent stromal cells produce and secret a multitude of small molecules including cytokines, growth factors, and ECM components, which can create an immunosuppressive, inflammatory TME that may promote tumor growth and metastasis [24]. Among those secreted small molecules, matrix metalloproteinases (MMPs) modify the ECM, contributing to premature stroma aging, and are involved in angiogenesis, which promotes cancer cell growth and migration [25]. Another important factor in TME is IL-8, which is a chemokine with various pro-tumorigenic functions within the TME. It promotes tumor cell proliferation and transformation into migratory or mesenchymal phenotypes, angiogenesis, and the recruitment of immunosuppressive cells [26].

Against this background, we investigated the expression of five surrogate markers of T cell exhaustion (PD-1, TIM-3, LAG-3) and stromal senescence (IL-8, MMP-3) in a cohort of *n* = 116 HNSCC patients. By correlating the biomarker expression with the patients’ clinical and histopathological data as well as vitamin D status, we aimed to gain new insights into head and neck cancer tumor and immunological biology as well as potentially identify new therapeutic strategies targeting the tumor microenvironment.

## 2. Results

### 2.1. Expression of T Cell Exhaustion and Stromal Senescence Markers Correlates with HPV Tumor Status but Not Vitamin D Serum Level

In an initial step, a semiquantitative analysis of intratumoral immunohistochemical staining targeting the five biomarkers PD-1, TIM-3, LAG-3, IL8, and MMP-3 was performed, followed by an assessment of peritumoral staining. The analysis revealed distinct differences in expression levels, ranging from negative or minimal staining (score 0–1) to high levels of staining (score up to 12). A representative illustration is provided in Figure 1, where, on the left, low expression levels are demonstrated by a corresponding low number of positively stained cells and a weak staining intensity. On the right, tumors and the TME exhibit a significantly stronger staining intensity and a higher number of positively stained cells. A subsequent analysis of immune cell infiltration patterns with respective IHC biomarker staining in the peritumoral and intratumoral region is presented in Figure 2, highlighting the variability in the degree of infiltration across the tumor microenvironment. The intratumoral and peritumoral biomarker expression was correlated with the HPV tumor status (positivity defined as p16+ and HPV-DNA+), as well as the 25-OH-vitamin D serum level (VitD low ≤ 10 ng/mL; VitD high > 10 ng/mL).

Here, HPV-positive tumors showed a significant increase in both the peritumoral and intratumoral expression of the T cell exhaustion markers PD-1 (*p* = 0.0142, peritumoral median IRS of 0.0 vs. 0.5; *p* = 0.0344, intratumoral median IRS of 0.0 vs. 0.0) and TIM-3 (*p* = 0.0180, peritumoral median IRS of 6.0 vs. 3.5; *p* = 0.0315, intratumoral median IRS of 4.0 vs. 2.5) as shown in Figure 2. The LAG-3 expression demonstrates a clear tendency toward increased levels in association with HPV infection; however, statistical significance was not reached, with *p* = 0.21 for peritumoral and *p* = 0.12 for intratumoral regions (median IRS of 2.0 vs. 1.0 peritumoral and 2.0 vs. 1.0 intratumoral). By contrast, HPV infection resulted in a highly significant suppression (*p* = 0.0076) of MMP-3 expression on intratumoral TILs (median IRS of 1.5 vs. 3.0), while the expression on peritumoral TILs remained unchanged. No influence of the HPV tumor status on the fibroblast expression of stroma senescence surrogate markers could be observed. Regarding the patients’ vitamin D status, none of the analyzed biomarkers showed a significant correlation with vitamin D supply. However, there was a trend toward increased expression levels of PD-1, TIM-3, and LAG-3, and a decreased expression level of MMP-3 in VitD-high patients compared to VitD-low patients. For LAG-3, the intratumoral median IRS was 1.0 in both groups, while the peritumoral median IRS was 1.5 in VitD-low and 1.0 in VitD-high patients. TIM-3 showed intratumoral median IRS values of 2.5 for VitD-low and 3.0 for VitD-high patients, with a consistent peritumoral IRS of 3.5 in both groups. For PD-1, the intratumoral median IRS was 2.0 in VitD-low patients and 3.0 in VitD-high patients, while the peritumoral IRS was 1.5 in both groups. The IL-8 expression showed no variation, with intratumoral and peritumoral IRS values of 2.0 and 1.5, respectively, for both groups. Lastly, MMP-3 had a higher intratumoral IRS in VitD-low patients (3.0) compared to VitD-high patients (2.0), with identical peritumoral IRS values of 2.5 for both groups. No significant differences were observed across all biomarkers analyzed.

### 2.2. High Levels of PD-1, LAG-3, and TIM-3 Expression Predict Improved Overall Patient Survival

Looking at the potentially prognostic relevance of the analyzed surrogate markers of T cell exhaustion and stroma senescence, we found significant correlations only for the T cell exhaustion markers PD-1, TIM-3, and LAG-3. Here, a high expression of PD-1 (defined by a PD-1 expression above the statistical mean of all samples) correlated with a significantly improved overall survival. PD-1-high patients showed a two-year overall survival of 87% in contrast to 58% in the PD-1-low group. This correlation was significant for PD-1 expression on both peritumoral (*p* = 0.0101) and intratumoral immune cells (*p* = 0.0266, Figure 3E,F).

A similar trend was observed for TIM-3: a high peritumoral TIM-3 expression showed a highly significant (*p* < 0.0001) overall survival benefit compared to low peritumoral TIM-3 expression. The two-year overall survival within these groups was 86% (TIM-3 peritumoral high) vs. 51% (TIM-3 peritumoral low). For TIM-3 expression on intratumoral immune cells, comparable effects were shown. Moreover, a significant survival advantage was observed for highly expressed LAG-3 on peritumoral TILs with a two-year survival of 77% (LAG-3 peritumoral high) vs. 56% (LAG-3 peritumoral low). However, no significant overall survival advantage was observed for a high intratumoral LAG-3 expression (*p* = 0.34).

By contrast, there was no significant correlation with the patients’ overall survival regarding the expression of the stromal senescent markers MMP-3 and IL-8 on intra- and peritumoral fibroblasts (Figure 3G–J).

### 2.3. Positive HPV Tumor Status but Not Vitamin D Serum Level Predicts Improved Overall Patient Survival

The overall survival of the included HNSCC patients was correlated with the HPV tumor status and vitamin D status. We could show that HPV-positive HNSCCs showed a significantly better prognosis, with an overall survival rate of 85% after 2 years, compared to 55% in HPV-negative cases. For vitamin D, a trend towards improved overall survival in VitD-high patients compared to VitD-low patients was observed, particularly within the first 24 months after diagnosis (Figure 4). However, no statistical significance could be achieved.

## 3. Discussion

HNSCCs are among the most common cancers worldwide, presenting a substantial social and economic burden [1]. Advanced-stage HNSCCs have limited treatment options and are associated with a poor prognosis [3]. Immune checkpoint inhibitors (ICIs) such as the PD-1/PD-L1 axis blockade provide new therapeutic options for recurrent or metastatic HNSCCs, but response rates remain modest with clinically relevant remissions being observed in less than 25% of cases [9,27]. Hence, new immunotherapeutic strategies are urgently needed and have been intensively studied over the past years with an increasing focus on the immunological and non-immunological tumor microenvironment [28]. The PD-1 pathway enables tumor cells to evade immune surveillance and resist treatment [29]. Anti-PD-1/PD-L1 antibodies have shown promise as checkpoint inhibitors, with overall low response rates, and adverse events have been noted, underscoring the need for a better understanding of the PD-1-mediated immunosuppression of cancer [29].

In this context, we investigated the expression of the T cell exhaustion markers PD-1, TIM-3, and LAG-3 on intra- and peritumoral TILs as well as the expression of the stroma senescence markers IL-8 and MMP-3 on intra- and peritumoral tumor-associated fibroblasts in a cohort of n = 116 HNSCC patients. We found a significant survival benefit for patients with an increased expression of the T cell exhaustion markers PD-1, LAG-3, and TIM-3 on intra- and peritumoral immune cells while the expression of the stroma senescence markers MMP-3 and IL-8 on intra- and peritumoral fibroblasts showed no influence on the patient outcome. 

Considering the prognostic value of the aforementioned biomarkers in head and neck cancer, current literature evidence remains sparse with only a few studies including in most cases only a limited subset of patients.

The immune markers LAG-3, TIM-3, and PD-1 were examined in a multicenter study by Zou et al. in head and neck lymphoepithelioma-like carcinomas [30]. TIM-3 and LAG-3 were co-expressed with markers like PD-L1, B7H3, IDO-1, and VISTA, indicating a role in immune regulation within the tumor microenvironment. However, high expressions of these biomarkers were linked to worse disease-free and overall survival. The contrast to our findings may be linked to the relationship between checkpoint inhibitor expression and HPV infection. In HPV-positive HNSCCs, a higher expression of markers like TIM-3 and LAG-3 suggests a more active immune environment, potentially leading to a better response to checkpoint blockade therapy. HPV-positive tumors typically show greater immune cell infiltration, including TILs, which express these markers [31]. Conversely, HPV-negative tumors often have a less active immune landscape and respond poorly to immunotherapy. Thus, in HPV-positive cases, elevated checkpoint expression may signal a better therapeutic prognosis due to increased immunogenicity.

Another study from Yang et al. found that TIM-3 was highly expressed on intratumoral and/or stromal TILs in 91.3% of HNSCC cases [32]. TIM-3 TIL expression correlated with the tumor size, lymph node metastasis, and TNM stage, with lower TIM-3+ TIL levels linked to significantly better survival and prognosis. Here, too, there are discrepant results to our trials, but these can be attributed to the positivity of the HPV status. The results suggest that TIM-3 is a potential oncologic target in HNSCCs.

When looking at the stromal senescence markers IL-8 and MMP-3, no significant effect on overall survival was observed, but there was a significant association of MMP-3 expression with HPV positivity. Liu et al. could show that MMP-3 mRNA expression was elevated in HNSCCs compared to normal tissue and was significantly correlated with the pathological stage of HNSCC patients [33]. Additionally, MMP-3 expression correlated with immune cell infiltration, and as significant predictors of clinical outcomes in HNSCCs.

With respect to the HPV tumor status, the significantly improved overall survival in HPV-positive compared to HPV-negative HNSCC patients in our cohort is in line with numerous prospective and retrospective large-scale clinical studies of the past years and underlines the outstanding relevance of HPV as a prognostic and predictive biomarker in head and neck cancer, especially in oropharyngeal SCCs [34,35]. Considering the relevance of HPV for the response to immune checkpoint inhibition, phase III clinical trials that led to the FDA and EMA approval of pembrolizumab and nivolumab for RM-HNSCC treatment found no predictive value of HPV. However, several studies have shown better outcomes of HPV+ HNSCC patients undergoing a PD-1/PD-L1 axis blockade in contrast to HPV- HNSCC patients [36]. Exemplarily, Wang et al. demonstrated that the HPV status can predict the efficacy of PD-1 inhibition in HNSCC patients independent of PD-L1 expression, likely due to an HPV-induced inflamed immune microenvironment [37]. As our study showed that a positive HPV tumor status is associated with an increased expression of the T cell exhaustion biomarkers PD-1, TIM-3, and LAG-3, our results provide a potential explanation for their observation. ICI could thus positively influence and reactivate antitumoral T cell response, which seems to be driven into an exhausted stage in a much stronger manner than in non-HPV associated cases.

Apart from their relevance as potential prognostic biomarkers, the proteins investigated in our study may also serve as potential targets for new TME-directed immunotherapeutic strategies. Wuerdemann et al. demonstrated that intratumoral CD8+ T cells in oral HNSCCs showed a significantly upregulated expression of LAG-3, TIM-3, and VISTA, and concluded that those proteins could be used as targets for new immunotherapeutic strategies [38]. Indeed, numerous ongoing clinical trials are investigating the efficacy of ICIs targeting LAG-3 and TIM-3 in various cancer types including head and neck cancer (e.g., NCT04811027, NCT05287113), especially in combination with PD-1 antibodies. In melanoma, the dual LAG-3 and PD-1 inhibitor Opdualag was already approved by the FDA for treating unresectable or metastatic disease in adults and children [39,40]. In addition, combining ICIs or using bispecific antibodies (BsAbs) that target two ICPs at the same time is a promising approach to overcoming resistance to single-agent therapy as proven by the recently approved BsAbs targeting anti-LAG-3/TIGIT [40]. However, no LAG-3 and/or TIM-3 inhibitors are approved for the treatment of head and neck cancer so far, so further clinical studies are needed.

In addition to the HPV tumor status, we also investigated a potential correlation of the patients’ 25-OH-vitamin D serum level with T cell exhaustion and stroma senescence biomarkers. In previous projects, vitamin D was shown to stimulate the infiltration of TME in head and neck cancer with various immune cells subtypes and additionally enhance their anti-tumor effector function, resulting in improved patient survival [11]. In the present study, we only found a tendency towards improved overall survival in VitD-high patients, which is in line with numerous previous studies of our own and other groups [11,12,41,42,43]. However, we did not find any significant correlation of 25-OH-vitamin D serum levels with the expression levels of the T cell exhaustion and stroma senescence biomarkers investigated, suggesting that vitamin D has no major role in those molecular processes.

From a critical point of view, there are some limitations that need to be considered when interpreting the study results. Our study highlights the complex interplay of T cell exhaustion and stromal senescence markers within the tumor microenvironment and their potential prognostic value in HNSCC. While we observed associations between certain checkpoint markers and patient survival, the limited size and heterogeneity of our patient cohort constrain the generalizability of these findings. Additionally, the surrogate markers we used only partially represent the biology of T cell exhaustion and stromal senescence, as these markers are also involved in other cellular functions.

Future studies should leverage RNA sequencing and pathway analyses to provide a more comprehensive and specific assessment of these biological processes. Moreover, further analyses with a larger and more homogeneous cohort, ideally with multivariate models, would be crucial to confirm the independent prognostic relevance of these markers. Expanding upon these findings could guide the development of therapeutic approaches that more effectively target the unique immune landscape in HNSCCs.

Another limitation of this study is potential interobserver variability in IHC interpretation, despite involving three independent investigators, including a board-certified pathologist. While using the mean IRS reduced variability, standardized protocols or automated tools are needed for greater consistency in future studies.

In our study, we excluded salivary gland tumors, but their differential diagnosis is important. Salivary gland tumors, such as pleomorphic adenomas, Warthin’s tumors, and malignancies like mucoepidermoid carcinoma and adenoidcystic carcinoma, share clinical features with HNSCCs, complicating diagnosis. Accurate differentiation relies on clinical presentation, imaging, histopathology, and molecular profiling. Immunohistochemical markers like p63, CK7, and EGFR distinguish these tumors [44]. Molecular characteristics, such as MAML2 fusions in mucoepidermoid carcinoma, differentiate salivary gland tumors from HNSCCs, which often involve TP53 mutations [44]. Although salivary gland tumors were not included, further research into their molecular profiles and comparisons to HNSCC will enhance diagnostic accuracy and treatment strategies, improving patient outcomes.

## 4. Materials and Methods

A total of n = 116 patients with histologically proven HNSCC were included in our study. All patients were diagnosed and treated between 2006 and 2021 at the Department of Otorhinolaryngology, Head and Neck Surgery at the Saarland University Medical Center (Homburg/Saar, Germany). The patient cohort consisted of 97 male and 19 female patients with a median age of 64.2 years. Tumor node metastasis (TNM) and American Joint Committee on Cancer (AJCC) stages were defined according to the seventh version of the AJCC/Union for International Cancer Control (UICC) head and neck cancer staging system. Further epidemiological and clinical characteristics are shown in Table 1. The findings in this study are based on treatment modalities, specifically surgery, radiotherapy (RT), and radiochemotherapy (R(C)T).

For all patients, the pre-therapeutic 25-OH-vitamin D serum level was available and therefore included in our analyses. Here, a distinction was made between patients with a sufficient vitamin D supply (57 patients, 25-OH-vitamin D, ≥10 ng/mL, VitD-high) and insufficient vitamin D supply (59 patients, <10 ng/mL, VitD-low). All patients gave their written informed consent for the scientific use of their tissue samples and clinical data. All experiments were performed in accordance with the Declaration of Helsinki and its later amendments as well as the relevant guidelines and regulations. The study was approved by the Saarland Ethics Review Committee (reference number 280/10). For the experiments in our study, tumor tissue samples either taken during diagnostic panendoscopy for the histological verification of tumor diagnosis or during therapeutic tumor resection were used.

### 4.1. HPV Tumor Status

The HPV tumor status was determined using a combination of p16 immunohistochemical (IHC) staining and HPV-DNA-PCR analysis. Among the 116 HNSCCs patients, 81% were found to be HPV-negative, while 19% tested positive for HPV. Only those patients who showed both positive p16 IHC staining and positive HPV-DNA-PCR results were classified as having HPV-positive tumors. Due to the notably poorer prognosis and distinct tumor biology observed in discordant cases (where patients tested as p16-negative/HPV-positive or p16-positive/HPV-negative), it was predefined that both tests need to be positive to assign an HPV-positive tumor status.

For HPV-DNA-PCR testing, DNA was extracted from fresh-frozen tumor samples using the QIAamp DNA Blood Mini Kit (Qiagen, Hilden, Germany) according to the manufacturer’s protocol. The HPV-DNA-PCR was conducted on the LightCycler 2.0 (Roche Diagnostics, Mannheim, Germany) using GP5+/6+ primers, following previously established procedures. Detection of the PCR amplification products was achieved with SYBR Green and gel electrophoresis. The PCR process included an initial denaturation step at 95 °C for 15 min, followed by 45 cycles of denaturation at 95 °C for 10 s, annealing at 45 °C for 5 s, and elongation at 72 °C for 18 s. After amplification, a melting curve analysis was performed over a temperature range of 45 °C to 95 °C, with an increase of 0.2 °C per second. Each PCR run included HPV16- and HPV18-positive controls, with melting temperatures (Tm) of 79 °C and 82 °C, respectively. The gene for glyceraldehyde-3-phosphate dehydrogenase (GAPDH) was amplified as an internal control.

For the immunohistochemical detection of p16, the CINtec p16 histology kit (Roche Diagnostics) was used according to the manufacturer’s guidelines on formalin-fixed paraffin-embedded tissue samples obtained as described below. Epitope retrieval was achieved by heat-induced unmasking after deparaffinization in a rice cooker for 20 min, using the provided retrieval buffer. The p16 antibody was then applied, and the detection of staining was performed as recommended. Each batch of staining included both positive and negative controls.

### 4.2. Immunohistochemistry

Formalin-fixed paraffin-embedded (FFPE) tissue samples of the included patients were used for histopathological and immunohistochemical analyses of the tumor microenvironment (TME). Therefore, fresh tissue samples were first placed in PBS-buffered 4% formalin for 24 h and then embedded in paraffin using Tissue-Tek^®^VIPTM5 JR (Olympus, Tokyo, Japan). Next, FFPE tissue sections were prepared to perform immunohistochemical staining targeting PD-1, TIM-3, IL-8, LAG-3, and MMP-3. Initially, 3 tissue sections of 10 μm thickness were discarded to subsequently generate 3 μm thick sections using a Leica RM2235 rotary microtome (Leica Microsystems, Wetzlar, Germany). Sections were then transferred onto Superfrost Ultra Plus microscope slides (Menzel-Gläser, Braunschweig, Germany) and dried at 37 °C overnight. Deparaffinization was carried out, followed by heat-induced epitope unmasking in a 10 mM citrate buffer (pH 6.0). Nonspecific binding sites were blocked by the subsequent incubation of the slides with 3% BSA (Sigma Aldrich, St. Louis, MO, USA) in PBS (Sigma Aldrich) at pH 7.2 for 30 min. Sections were then exposed to primary antibodies targeting IL-8 (1:2350, ab18672), PD-1 (1:750, ab52587), TIM-3 (1:5600, ab241332), LAG-3 (1:350, ab209236), and MMP-3 (1:1000, ab52915; all antibodies from abcam, Cambridge, UK) for 1 h at room temperature. Visualization was performed using the Dako REALTM Detection System, Alkaline Phosphatase/RED (Dako Agilent Technologies, Glostrup, Denmark) according to the manufacturer’s instructions. Finally, counterstaining with hematoxylin (Sigma Aldrich) was performed before the slides were mounted with coverslips.

A semiquantitative analysis of immunohistochemically stained tumor samples was performed using the Immunoreactivity Score (IRS) according to Remmele and Stegner (1987). The IRS assigns numerical values from 0 to 4, depending on the staining intensity and the percentage of stained cells in relation to all cells. The grading includes no reaction (0), weak staining reaction (1), moderate staining reaction (2), and strong staining reaction (3). The percentage of stained cells in relation to all cells was quantified with 0% (0), <10% (1), 10–50% (2), 51–80% (3), and >80% (4). Both numerical values were then multiplied resulting in a final IRS between 0 (negative) and 12 (strongly positive). For PD-1, TIM-3, and LAG-3, only immunoreactivity on peri- and intratumoral leukocytes was analyzed, and for MMP-3 and IL-8, only immunoreactivity on intra- and peritumoral fibroblasts was analyzed. The boundary between intratumoral and peritumoral regions was defined based on histopathological landmarks. The intratumoral region refers to the area within the tumor mass, including tumor cell nests and the immediately surrounding stroma infiltrated by leukocytes. By contrast, the peritumoral region is defined as the stromal area within a close margin around the tumor mass, carefully avoiding overlap with adjacent non-tumor tissues. To distinguish CAFs microscopically, we relied on their characteristic spindle-shaped morphology, elongated nuclei, and spatial localization within the stromal compartment of the tumor. All IHC stainings were analyzed by three independent investigators including one board-certified pathologist. For statistical analyses, the arithmetic mean of the three IRS values per tissue sample was used. To define whether a high IHC expression was present, the mean IRS of all samples was determined. The individual samples were defined as having either high or low expression according to the mean value as a diagnostic threshold.

### 4.3. Statistical Analysis

For statistical analysis, Prism 9 software (GraphPad Software, Boston, MA, USA) was used. To check the acquired data for Gaussian distribution, the Anderson–Darling test, D’Agostino and Pearson test, Shapiro–Wilk test, and Kolmogorov–Smirnov test were used. If data passed ≥2 of the normality tests, parametric tests were used for statistical testing (unpaired *t* test with Welch’s correction, one-way ANOVA test). If the data did not pass ≥2 of the aforementioned normality tests, non-parametric tests were used (Mann–Whitney U test, Kruskal–Wallis test). The overall survival rates of the patient collective were analyzed using the Mantel–Cox test (log-rank test) and presented in Kaplan–Meier curves. *p* values < 0.05 were considered statistically significant (α = 0.05). The tests that were used for statistical testing are indicated in the figure legends or the text, respectively.

## 5. Conclusions

Taken together, we have shown that the increased expression of the T cell exhaustion markers PD-1, LAG-3, and TIM-3 is associated with a significantly improved overall survival in HNSCC patients, and that HPV-positive disease is associated with an increased expression of these biomarkers. Further studies are necessary to uncover the clinical relevance of these observations and evaluate a potential clinical use of T cell exhaustion markers as single or combinational immunotherapeutic targets in head and neck cancer therapy.

## Figures and Tables

**Figure 1 ijms-25-13490-f001:**
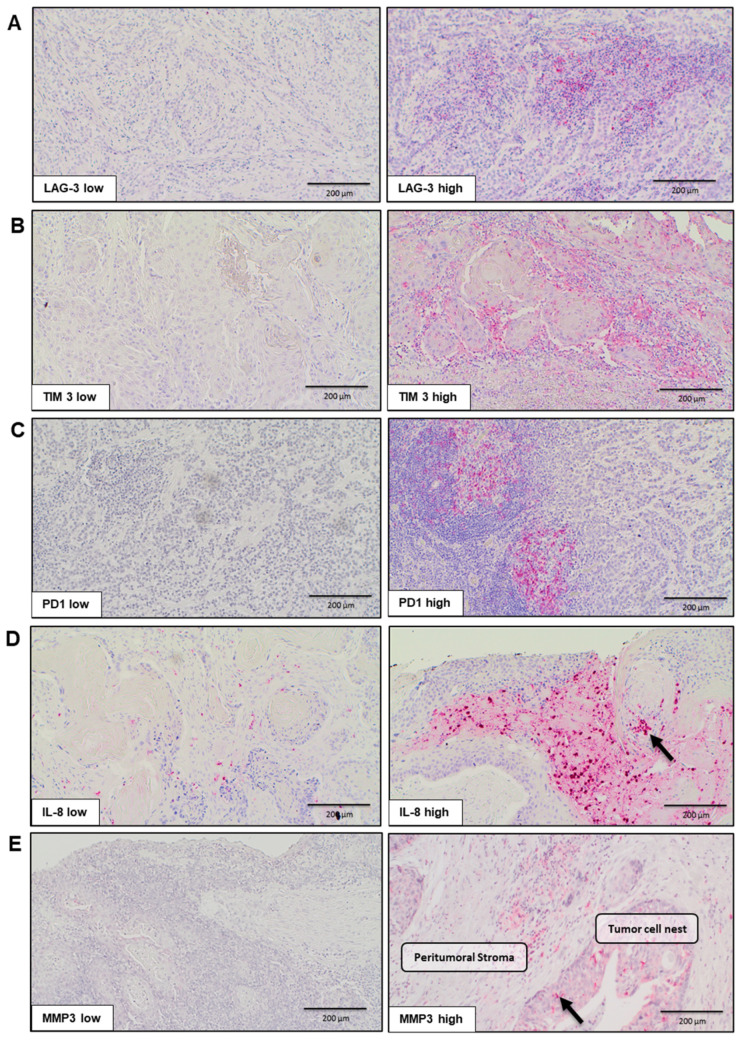
IHC representation of immune markers LAG-3, TIM-3, PD-1, IL-8, and MMP-3 (**A**–**E**). The images illustrate representative sections of the tumor as well as the tumor microenvironment. On the left side, a low immune reactive score (IRS) is depicted, characterized by a small number of positively stained cells and a weak staining intensity. On the right, a correspondingly high IRS is shown. In panels D and E, arrows indicate positively stained cells. The intratumoral regions are identified by tumor cell nests, while the peritumoral stroma is shown accordingly. Magnification: 10×.

**Figure 2 ijms-25-13490-f002:**
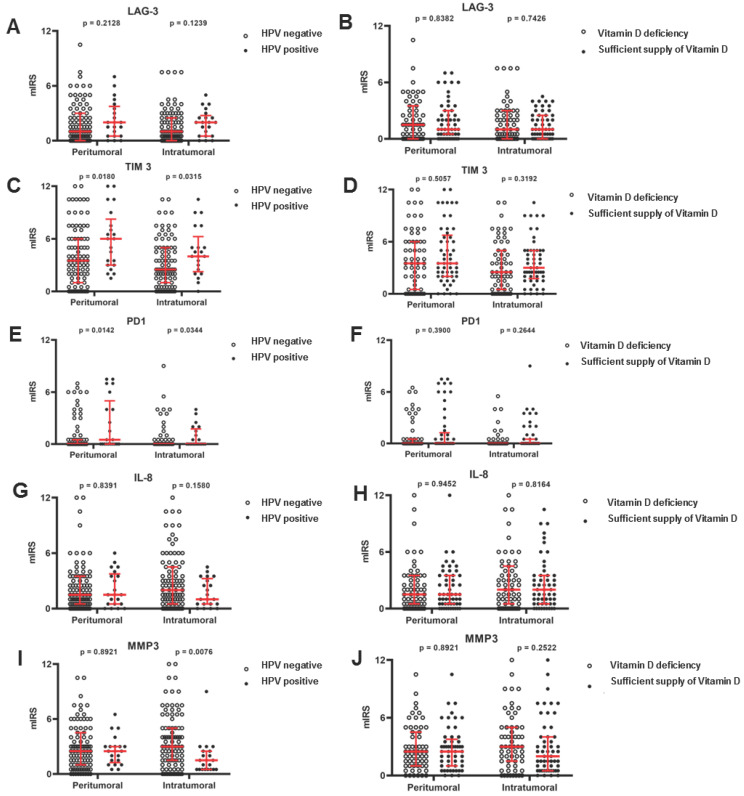
Correlation of the expression of T cell exhaustion and stroma senescence biomarkers with HPV tumor status and vitamin D status. (**A**,**B**) IRS of LAG-3 depending on HPV and vitamin D status. (**C**,**D**) IRS of TIM-3 depending on HPV and vitamin D status. (**E**,**F**) IRS of PD-1 depending on HPV and vitamin D status. (**G**,**H**) IRS of IL-8 depending on HPV and vitamin D status. (**I**,**J**) IRS of MMP-3 depending on HPV and vitamin D status. Statistical analysis was performed using the Mann–Whitney U test in all cases. The black dots symbolize one patient each, and the red lines show the median with the interquartile range.

**Figure 3 ijms-25-13490-f003:**
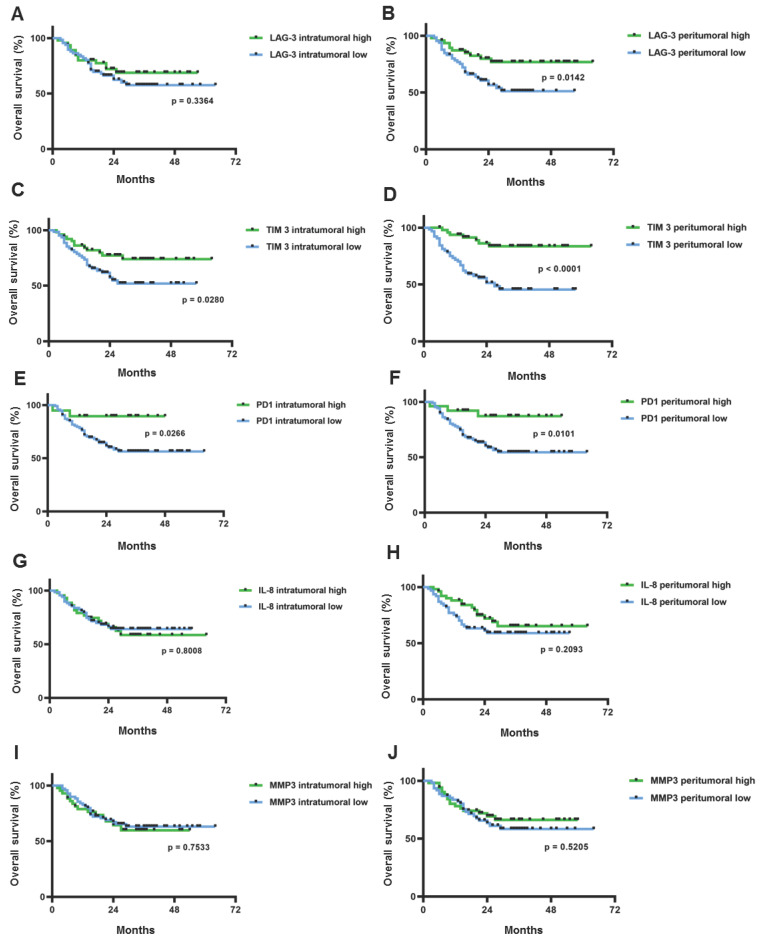
Overall survival of HNSCC patients depending on intratumoral and peritumoral expression of T cell exhaustion and stroma senescent biomarkers. Patient overall survival depending on LAG-3 expression (**A**,**B**), TIM-3 expression (**C**,**D**), and PD-1 expression (**E**,**F**) on intratumoral (left image) and peritumoral TILs (right image), respectively. From (**G**–**J**), the influence of intratumoral (left image) and peritumoral (right image) expression of stroma senescence markers on tumor-associated fibroblasts on overall survival is shown (**G**,**H**) for IL-8; (**I**,**J**) for MMP-3. The log-rank test (Mantel–Cox) was used for the statistical analysis in each case. Censored data are indicated as black dots on the Kaplan–Meier curves.

**Figure 4 ijms-25-13490-f004:**
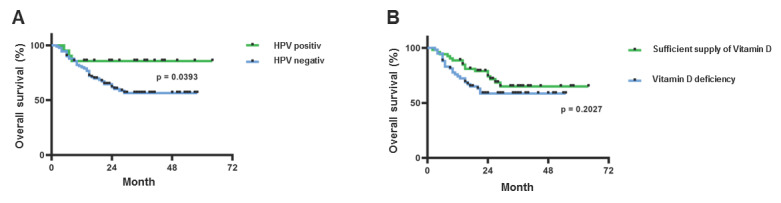
Patients’ overall survival depending on HPV tumor status and vitamin D status. (**A**) Illustration of overall survival as a function of HPV tumor status. (**B**) Illustration of overall survival as a function of vitamin D status. The log-rank test (Mantel–Cox) was used for the statistical analysis in each case. Censored data are indicated as black dots on the Kaplan–Meier curves.

**Table 1 ijms-25-13490-t001:** Demographic and clinical patient data. * The 7th version of the TNM/UICC classification was used to categorize the carcinomas.

		HNSCC Patients
No. of Patients		116
**Sex**	male	97 (84%)
	female	19 (16%)
**Median age [years]**	male	64.5
	female	62.8
**HPV Status**	positive	22 (19%)
	negative	94 (81%)
**Vitamin D Status**	high	57 (49%)
	low	59 (51%)
**Primary tumor**	oral cavity	36 (31%)
	larynx	32 (28%)
	oropharynx	31 (27%)
	hypopharynx	11 (9%)
	multiple localizations	6 (5%)
**T * stage**	1	18 (16%)
	2	48 (41%)
	3	26 (22%)
	4	24 (21%)
**N * stage**	0	38 (32%)
	1	18 (16%)
	2	52 (45%)
	3	8 (7%)
**M * stage**	0	108 (93%)
	1	8 (7%)
**UICC * Stage**	I	14 (12%)
	II	14 (12%)
	III	24 (21%)
	IVa	49 (42%)
	IVb	7 (6%)
	IVc	8 (7%)

## Data Availability

The raw data supporting the conclusions of this article will be made available by the authors upon request.

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
