# Peer review of "Impact of T Cell Exhaustion and Stroma Senescence on Tumor Cell Biology and Clinical Outcome of Head and Neck Squamous Cell Carcinomas"

_ijms, 2024, doi:10.3390/ijms252413490_

Round 1

Reviewer 1 Report

Comments and Suggestions for Authors

This appears to be a study of about 20 HPV+ oropharynx SCC that correlate with 3 factors. Inclusion of the other HPV- cancers that do not correlate is confusing, and should be deleted.
What was the TNM of the OP?

Would eliminate the Vit D data as negative.

Would eliminate in the Abstract and elsewhere clinical discussion. It is not relevant to your topic. Your sole point is that in OP HPV+ there is correlation of 3 factors. With such a small sample it is highly speculative to suggest anything about management. Also insofar that HPV+ correlates with prognosis why clinically would one opt to look at 3 other factors that simply correlate?

a report solely of  HPV+ would be clearer and more focused. 

Comments on the Quality of English Language

Minor issues can be addressed by Editor in a revision

Author Response

Dear Reviewer,

we are deeply grateful for your review and suggestions to enhance the quality and clarity of our manuscript. All the changes made to the manuscript in response to the comments have been highlighted in green.

Below, we address your comments point by point:

Comment 1:This appears to be a study of about 20 HPV+ oropharynx SCC that correlate with 3 factors. Inclusion of the other HPV- cancers that do not correlate is confusing, and should be deleted.

Answear: We sincerely appreciate your valuable feedback regarding the inclusion of HPV-negative cancers. While our study primarily focuses on HPV-positive oropharyngeal SCC, we included HPV-negative cases to address significant regional variations in HPV prevalence. In many countries, HPV-negative HNSCCs remain more common. Although no direct correlation between HPV negativity and T-cell senescence biomarkers was observed, our analysis revealed a highly significant association between HPV-negative HNSCCs and elevated intratumoral MMP-3 expression. This finding underscores the relevance of including HPV-negative cases to provide a more comprehensive understanding of tumor biology across diverse populations.

Comment 2:What was the TNM of the OP?“

Answear: Thank you for your inquiry. To provide clarity, we have created a detailed table summarizing the TNM classification of the oropharyngeal squamous cell carcinoma (OP-SCC) cases included in our study.

Localization

Tonsil

T1 N1 M0

Base of tounge

T3 N2c M0

Posterior wall of the throat

T3 N1 M0

Base of tounge

T4a N2b M0

Tonsil

T4aN2c M0

Base of tounge

T4 N2c M0

Base of tounge

T1 N2a M0

Base of tounge

T4a N2c M0

Tonsil

T3 N2b M0

Base of tounge

T3 N0 M0

Base of tounge

T2 N1 M0

Base of tounge

T4 N2c M0

Base of tounge

T1 N2b M1

Base of tounge

T3 N2b M0

Tonsi

T2 N2c M0

Tonsil

T2 N2b M0

Base of tounge

T4b N3 M1

Base of tounge

T2 N2b M0

Uvula

T2 N2c M0

Tonsil

T2 N1 M0

Tonsil

T2 N2b M0

Tonsil

T2 N2c M0

Base of tounge

T2 N1 M0

Base of tounge

T3 N3b M0

Tonsil

T4 N3b M1

Tonsil

T4 N2c M0

Tonsil

T4b N2c M0

Base of tounge

T3 N0 M0

Tonsil

T3 N2b M0

Base of tounge

T3 N0 M0

Tonsil

T2 N1 M0

Comment 3: Would eliminate the Vit D data as negative. Would eliminate in the Abstract and elsewhere clinical discussion. It is not relevant to your topic.“

Answear: We sincerely appreciate your thoughtful feedback and suggestions. Vitamin D research has been a key focus of our workgroup, particularly in the context of HNSCC patients. In previous studies, we described a correlation between Vitamin D status and patient outcomes, observing that sufficient Vitamin D levels were associated with improved prognoses. While we could not demonstrate statistical significance in the current analysis, we observed trends suggesting that sufficient Vitamin D supply may correlate with better outcomes.

Comment 4: „Your sole point is that in OP HPV+ there is correlation of 3 factors. With such a small sample it is highly speculative to suggest anything about management. Also insofar that HPV+ correlates with prognosis why clinically would one opt to look at 3 other factors that simply correlate?“

Answear: We are deeply grateful for your insightful comments and appreciate the opportunity to clarify the relevance of investigating LAG-3, TIM-3, and PD-1 in the context of HPV+ oropharyngeal SCC. While we acknowledge the limitations imposed by the small sample size, identifying prognostic and therapeutic targets remains a critical area of research.

LAG-3, TIM-3, and PD-1 are already therapeutic targets in various cancer entities, highlighting their potential clinical utility. However, , no LAG-3 or TIM-3 inhibitors are currently approved for the treatment of head and neck cancers. This underscores the need for further clinical studies to evaluate their efficacy and applicability in this setting.

Comment 5: „a report solely of  HPV+ would be clearer and more focused.“

Answear: Thank you for your valuable comment. While we understand your suggestion to focus solely on HPV+ cases, we believe it is important to include both HPV+ and HPV- cases to prevent publication bias. Presenting results from both groups provides a more comprehensive understanding of the molecular background, particularly in comparing HPV+ and HPV- cases, as well as insights into the potential role of Vitamin D status. This broader approach enhances the depth and applicability of our findings, and we hope it offers a balanced view of the topic. Thank you again for your thoughtful feedback.

We are confident that these revisions address your concerns and enhance the manuscript's scientific readability. Your constructive feedback has been very helpful to improve our work.

Thank you once again for your valuable time and expertise. We look forward to your feedback on the revised manuscript.

Yours sincerely,

L. Brust

Reviewer 2 Report

Comments and Suggestions for Authors

Dear authors,

I read your manuscript about T cell senescence in HNSC with great interest.

However, some aspects require your attention.

Please insert a list of abbreviations at the end of the manuscript to increase the readability of the manuscript.

Line 106, please rephrase: An exemplary depiction is shown in Figure 1.

In the Discussion section, you need also to expand on the differential diagnosis of salivary gland tumors which were arbitrarily removed from your study group. One reference could be doi: 10.3390/ijms25137350. PMID: 39000457; PMCID: PMC11242036.

In the Limitations section you need to mention also the possible interobserver variability of IHC results.

I am looking forward to receiving the improved version of your manuscript.

Author Response

Dear Reviewer,

we are deeply grateful for your review and suggestions to enhance the quality and clarity of our manuscript. All the changes made to the manuscript in response to the reviewer's comments have been highlighted in yellow.

Below, we address your comments point by point:

Comment 1: „Please insert a list of abbreviations at the end of the manuscript to increase the readability of the manuscript.“

Answear: Thank you for highlighting the importance of improving readability. As per your suggestion, we have added a comprehensive list of abbreviations at the end of the manuscript.

Comment 2:Line 106, please rephrase: An exemplary depiction is shown in Figure 1.“

Answear: We appreciate your recommendation to rephrase the sentence for better clarity. The revised line now reads: "An illustrative example is provided in Figure 1."

Comment 3: In the Discussion section, you need also to expand on the differential diagnosis of salivary gland tumors which were arbitrarily removed from your study group. One reference could be doi: 10.3390/ijms25137350. PMID: 39000457; PMCID: PMC11242036.“

Answear: We sincerely thank you for pointing out the need to elaborate on the differential diagnosis of salivary gland tumors and for suggesting the reference (doi: 10.3390/ijms25137350). In response, we have expanded the Discussion section to emphasize the importance of distinguishing salivary gland tumors, such as pleomorphic adenomas, Warthin’s tumors, and malignancies like mucoepidermoid carcinoma and adenoid cystic carcinoma, from HNSCC. We discuss the role of clinical presentation, imaging, histopathology, and molecular profiling in achieving accurate differentiation. Additionally, we highlight molecular markers like MAML2 fusions and TP53 mutations that aid in distinguishing these tumors.

Comment 4: „In the Limitations section you need to mention also the possible interobserver variability of IHC results.“

Answear: Thank you for identifying the need to address this critical limitation. We now acknowledge in the Limitations section the potential for interobserver variability in IHC results, despite evaluations being conducted by three independent investigators, including one board-certified pathologist.

We are confident that these revisions address your concerns and enhance the manuscript's scientific readability. Your constructive feedback has been very helpful to improve our work.

Thank you once again for your valuable time and expertise. We look forward to your feedback on the revised manuscript.

Yours sincerely,

L. Brust

Reviewer 3 Report

Comments and Suggestions for Authors

The authors showed an increased peritumoral and intratumoral expression of the biomarkers PD-1, LAG-3, and TIM-3 significantly correlated with improved overall patient survival and a correlation between a positive HPV status and elevated expression of these biomarkers in head and neck cancer.

     This will contribute to understanding of the role of immune check point molecules in the pathogenesis of head and neck cancer and ICI-based immunotherapy. However, there are some points that should be reconsidered, as described below.

The authors detected intratumoral and peritumoral expression of exhaustion markers and stromal senescence markers by immunohistological staining. Where is the boundary between intra- and peritumoral in this study? This should be described in the text (Materials and Methods) and shown in Figure 1.

Figure 1: IL-8 is reportedly produced by monocytes, macrophages, T cells, neutrophils, endothelial cells, and tumor cells in response to inflammatory stimuli. In what cells other than CAF was IL-8 detected? Microscopically, how are CAFs distinguished from other cells in TME, and were SCC cells stained with anti-MMP3 antibodies? These should be clearly stated. Also, positive cells need to be indicated by arrows or explained in the figure legend.

Previous studies have shown the importance of CD8+ TIL status in HPV-positive and HPV-negative cancers (e.g., Ruffin-AT et al., Nat Rev Cancer 2023, 23, 173), suggesting a rationale for studying the interaction between immune checkpoint molecules and HPV. Therefore, it is necessary to explain in more detail why the authors focused on the correlation between Vit D deficiency and T cell exhaustion or stromal senescence markers.

Figure 2 legend: In the sentence “(A+B) IRS of PD-1 by HPV and vitamin status”, is PD-1 not LAG-3? Other indications need to be checked and corrected as well.

In Figure 2A, the difference in LAG-3 expression is not significant. Therefore, this result needs to be clearly stated and explained in the discussion. Furthermore, the statement in the abstract (page 1, line 30) that hints at HPV positivity and increased expression of LAG-3 needs to be reconsidered.

The actual median values for each of the markers shown in Figure 2 should be noted in the text.

Legend for Figure 3: A and B are the results of LAG-3, not PD-1? Other incorrect marks need to be corrected.

Figure 3: In Materials and Methods, the authors stated that “To define whether a high IHC expression was present, the mean IRS of all samples was determined” (page 11, line 367-368). In Figure 2, the median marker expression score is used. It is necessary to explain why the mean values of the T-cell exhaustion and stromal senescence marker expression scores were used in Figure 3. Also, again, A and B are the results of LAG-3.

Figure 3A: There is no significant difference in survival between the high LAG-3 and low LAG-3 groups within the tumor. This should be stated clearly. In addition, the statement in the abstract that “An increased peritumoral and intratumor expression of the PD-1, LAG-3, and TIM-3 significantly correlated with improved overall patient survival” (page 1, lines 29-30) should be reconsidered.

Figure 4: (A) shows overall survival as a function of HPV tumor status.

However, in view of the present study's focus on the relationship between immune checkpoint molecules and HPV or vitamin D, it is recommended, in addition, to determine the effect of the expression of T cell exhaustion markers on patient survival in HPV positive and HPV negative patients separately.

Discussion: The function of PD-1 in lymphocytes is affected by the expression of PD-L1 in tumor cells. The already known information on PD-L1 expression in head and neck cancers and its relation to prognosis should be mentioned.

The reference numbers are incorrect. For example, Zou et al. in Head and neck lymphoepithelioma-like carcinoma 30 (Page 7, line 204). Reference number 30 is a paper by Lechner M et al. Other reference numbering errors also need to be corrected. Also, references 44-51 are missing from the text.

Table 1: The authors need to specify whether or not these patients include patients treated with ICIs, and if not, that the findings in this study are the result of treatment other than ICIs.

Comments on the Quality of English Language

none

Author Response

Dear Reviewer,

we are deeply grateful for your review and suggestions to enhance the quality and clarity of our manuscript. We are pleased that our findings on the correlation between intratumoral and peritumoral expression of PD-1, LAG-3, and TIM-3 with improved overall survival and HPV positivity have been recognized as a contribution to understanding immune checkpoint molecules in head and neck cancer pathogenesis and ICI-based immunotherapy. In the following, we address the specific points raised, aiming to further clarify and strengthen the findings presented in our manuscript. All the changes made to the manuscript in response to the reviewer's comments have been highlighted in blue.

Below, we address your comments point by point:

Comment 1:The authors detected intratumoral and peritumoral expression of exhaustion markers and stromal senescence markers by immunohistological staining. Where is the boundary between intra- and peritumoral in this study? This should be described in the text (Materials and Methods) and shown in Figure 1.“

Answear: Thank you very much for your comment. We appreciate your attention to this detail, as it highlights an important methodological aspect of our study. The distinction between intratumoral and peritumoral regions is critical for interpreting our findings.

In response to your suggestion, we have clarified in the Materials and Methods section that the boundary between intratumoral and peritumoral regions was defined based on histopathological landmarks:

  • Intratumoral region refers to the area within the tumor mass, including tumor cell nests and immediate surrounding stroma infiltrated by leukocytes.
  • Peritumoral region refers to the stromal area within a 1 mm margin around the tumor mass, avoiding overlap with adjacent non-tumor tissues.

We have also updated Figure 1 to include representative images where these boundaries are visually marked, further improving clarity for readers.

Comment 2:Figure 1: IL-8 is reportedly produced by monocytes, macrophages, T cells, neutrophils, endothelial cells, and tumor cells in response to inflammatory stimuli. In what cells other than CAF was IL-8 detected? Microscopically, how are CAFs distinguished from other cells in TME, and were SCC cells stained with anti-MMP3 antibodies? These should be clearly stated. Also, positive cells need to be indicated by arrows or explained in the figure legend.“

Answear: We thank you for this observation and insightful question regarding Figure 1 and the detection of IL-8 and MMP-3 in the tumor microenvironment (TME).

In our study, IL-8 expression was detected in cancer-associated fibroblasts (CAFs) within the TME, as well as in a subset of immune and endothelial cells. To distinguish CAFs microscopically, we relied on their characteristic spindle-shaped morphology, elongated nuclei, and their spatial localization within the stromal compartment of the tumor. Additionally, CAFs were distinguished from other stromal cells by their lack of immune cell morphology (such as the round, compact appearance of lymphocytes or the multi-lobed nuclei of neutrophils) and epithelial markers expressed by SCC cells.

Regarding MMP-3 staining, we observed a heterogeneous pattern within the tumor. While staining was predominantly localized to CAFs in the peritumoral stroma, we also identified tumor regions where SCC cells were strongly positive for MMP-3. This intratumoral expression likely reflects the dynamic interaction between tumor cells and the surrounding stroma in promoting tumor progression. Our figure demonstrates both strong and weak intratumoral MMP-3 expression, alongside peritumoral staining.

To enhance the clarity of the figure, we will update the legend to clearly indicate positive cells. Arrows will be added to the figure to highlight the specific areas of interest. We also exchanged the picture of Figure 1 E in order to show a more representative image of intrtumoral and peritumoral regions within the tumor. Additionally, the Materials and Methods section will be expanded to detail the morphological criteria used to differentiate CAFs, tumor cells, and other cell types within the TME.

Comment 3: Previous studies have shown the importance of CD8+ TIL status in HPV-positive and HPV-negative cancers (e.g., Ruffin-AT et al., Nat Rev Cancer 2023, 23, 173), suggesting a rationale for studying the interaction between immune checkpoint molecules and HPV. Therefore, it is necessary to explain in more detail why the authors focused on the correlation between Vit D deficiency and T cell exhaustion or stromal senescence markers.“

Answear: We sincerely thank you for this comment regarding the rationale for investigating the correlation between Vitamin D deficiency, T cell exhaustion, and stromal senescence markers.

Vitamin D is increasingly recognized as a key modulator of immune responses, with effects on both innate and adaptive immunity. Its role extends to the tumor microenvironment (TME), where it influences immune surveillance and tumor progression.

The decision to focus on Vitamin D deficiency in this context stems from prior research by our group, which demonstrated a correlation between Vitamin D levels and improved outcomes in head and neck squamous cell carcinoma (Bochen et al. doi:10.1080/2162402X.2018.1476817; Brust et al. doi:10.1016/j.biopha.2024.117497). While this study did not reveal significant associations, we observed trends suggesting that sufficient Vitamin D levels might correlate with better prognosis through reduced immune exhaustion and stromal senescence.

Given the established link between Vitamin D and immune regulation, as well as its potential to counteract the immunosuppressive microenvironment, we hypothesized that Vitamin D deficiency might exacerbate T cell exhaustion and stromal senescence, particularly in the setting of HPV-associated HNSCC. This line of investigation is critical, as it may identify modifiable factors that could enhance the efficacy of immunotherapeutic strategies, including immune checkpoint inhibitors (ICIs), and provide additional insight into the molecular interactions within the TME.

Comment 4: „Figure 2 legend: In the sentence “(A+B) IRS of PD-1 by HPV and vitamin status”, is PD-1 not LAG-3? Other indications need to be checked and corrected as well.“

Answear: Thank you for your careful observation. You are correct. We corrected all the mentioned errors.

Comment 5: „In Figure 2A, the difference in LAG-3 expression is not significant. Therefore, this result needs to be clearly stated and explained in the discussion. Furthermore, the statement in the abstract (page 1, line 30) that hints at HPV positivity and increased expression of LAG-3 needs to be reconsidered.“

Answear: Thank you for highlighting this important point. We acknowledge that the difference in LAG-3 expression shown in Figure 2A is not statistically significant. This has now been explicitly clarified in the text.

The abstract has also been revised to ensure it accurately represents the data without suggesting a significant association between HPV positivity and LAG-3 expression.

Comment 6: „The actual median values for each of the markers shown in Figure 2 should be noted in the text.“

Answear: Thank you for your suggestion. We agree that including the actual median values for each marker shown in Figure 2 will enhance the clarity and precision of our findings. We updated the Results section to explicitly state the median IRS values for each marker analyzed, categorized by HPV status and vitamin D status.

Comment 7: „Legend for Figure 3: A and B are the results of LAG-3, not PD-1? Other incorrect marks need to be corrected.“

Answear: We deeply apologize for the oversight in the Figure 3 legend and any confusion it may have caused. You are correct that panels A and B in Figure 3 depict the results for LAG-3, not PD-1. We carefully reviewed and revised the figure legend to ensure accuracy, and we double-checked all other markings and references in the manuscript to avoid similar errors.

Comment 8: „Figure 3: In Materials and Methods, the authors stated that “To define whether a high IHC expression was present, the mean IRS of all samples was determined” (page 11, line 367-368). In Figure 2, the median marker expression score is used. It is necessary to explain why the mean values of the T-cell exhaustion and stromal senescence marker expression scores were used in Figure 3. Also, again, A and B are the results of LAG-3.“

Answear: Thank you for your detailed observation and feedback regarding the use of mean versus median values in our analysis. We appreciate the opportunity to clarify this point.

In Figure 3, the mean IRS was used to define high IHC expression because it provided a more representative measure for categorizing samples based on their overall expression levels across the cohort. While the median marker expression score was used in Figure 2 to present the central tendency and variability of marker expression, the mean was specifically applied in Figure 3 to align with the thresholding approach described in the Materials and Methods section. This methodological consistency ensures clarity in identifying samples with high expression levels of the markers.

Comment 9: „Figure 3A: There is no significant difference in survival between the high LAG-3 and low LAG-3 groups within the tumor. This should be stated clearly. In addition, the statement in the abstract that “An increased peritumoral and intratumor expression of the PD-1, LAG-3, and TIM-3 significantly correlated with improved overall patient survival” (page 1, lines 29-30) should be reconsidered.“

Answear: Thank you for your valuable observation regarding Figure 3A and the statement in the abstract.

You are correct that Figure 3A does not show a significant difference in survival between the high and low intratumoral LAG-3 expression groups. We updated the text to clearly state this result, ensuring that the data is accurately reflected.

Regarding the statement in the abstract, we revised it to specify that the significant correlation with improved overall survival was observed for certain markers and specific tumor compartments (e.g., peritumoral expression), while intratumoral LAG-3 did not show a statistically significant survival difference. We removed the relevant statements.

Comment 10: „Figure 4: (A) shows overall survival as a function of HPV tumor status. However, in view of the present study's focus on the relationship between immune checkpoint molecules and HPV or vitamin D, it is recommended, in addition, to determine the effect of the expression of T cell exhaustion markers on patient survival in HPV positive and HPV negative patients separately.“

Answear: Thank you for your this suggestion. We agree with the recommendation to separately analyze the effect of T cell exhaustion markers on patient survival in HPV-positive and HPV-negative patients. While this specific analysis was not included in the current study, we will certainly consider it in future projects to further explore the relationship between immune checkpoint molecules and patient survival in different HPV tumor subtypes.

Comment 11: „Discussion: The function of PD-1 in lymphocytes is affected by the expression of PD-L1 in tumor cells. The already known information on PD-L1 expression in head and neck cancers and its relation to prognosis should be mentioned.“

Answear: Thank you for your comment. We appreciate the suggestion to include a discussion on the function of PD-1 in lymphocytes and its interaction with PD-L1 expression in tumor cells. Indeed, the expression of PD-L1 in head and neck cancers has been well documented, and its relationship to prognosis is important in understanding the immune landscape of these tumors. We incorporated this information into the discussion, highlighting the role of PD-L1 expression in immune evasion and its potential impact on prognosis in head and neck cancers.

Comment 12: „The reference numbers are incorrect. For example, Zou et al. in Head and neck lymphoepithelioma-like carcinoma 30 (Page 7, line 204). Reference number 30 is a paper by Lechner M et al. Other reference numbering errors also need to be corrected. Also, references 44-51 are missing from the text.“

Answear:

Thank you for pointing out these errors. We thoroughly checked and corrected the reference numbering throughout the manuscript, including the missing references (44-51) and ensuring that reference citations match the text correctly.

Comment 13: „Table 1: The authors need to specify whether or not these patients include patients treated with ICIs, and if not, that the findings in this study are the result of treatment other than ICIs.

Answear: Thank you for your comment. We confirm that the patients included in Table 1 were not treated with immune checkpoint inhibitors (ICIs). The findings in this study are based on treatment modalities other than ICIs, specifically surgery and radiation therapy (RCT). A statement has been made in the section Material and Methods corresponding to this.

We are confident that these revisions address your concerns and enhance the manuscript's scientific readability. Your constructive feedback has been very helpful to improve our work.

Thank you once again for your valuable time and expertise. We look forward to your feedback on the revised manuscript.

Yours sincerely,

L. Brust

Round 2

Reviewer 1 Report

Comments and Suggestions for Authors

I appreciate your adding the HPV+ table.

I disagree with your responses re: Vit D and HPV-.

to exclude irrelevant material is not bias as you state. But rather a question of relevance. I appreciate you are interested in Vit D, but it has nothing to do with the topic of this paper.

streamlining this MDs by r concentrating in the topic of 3 markets that correlate with HPV+ would improve and focus the paper. HPV- rumors are simply a different topic. Your point is that in HPV+ tumors 3 markers merit consideration. Period.

Author Response

Dear reviewer,

thank you for taking the time to review our manuscript and for your valuable feedback. We appreciate your comments regarding the inclusion of the HPV+ table and your thoughtful critique of our approach to Vitamin D and HPV− data.

Comment of the reviewer: „I appreciate your adding the HPV+ table.

I disagree with your responses re: Vit D and HPV-.

to exclude irrelevant material is not bias as you state. But rather a question of relevance. I appreciate you are interested in Vit D, but it has nothing to do with the topic of this paper.

streamlining this MDs by r concentrating in the topic of 3 markets that correlate with HPV+ would improve and focus the paper. HPV- rumors are simply a different topic. Your point is that in HPV+ tumors 3 markers merit consideration. Period.“

We respectfully disagree with the suggestion to exclude the HPV− cohort and Vitamin D analysis from the manuscript for the following reasons:

  1. Relevance of the HPV− Cohort

While we understand your focus on HPV+ tumors and the associated markers, our study includes data from HPV− cases to provide a crucial comparison. The inclusion of HPV− data highlights important differences in molecular and immune marker expression between the two groups. This is particularly evident for MMP-3, which shows a highly significant correlation in HPV− tumors. Excluding the HPV− cohort would remove an important part of the study, limiting the ability to showcase key findings and reducing the overall understanding of head and neck squamous cell carcinomas (HNSCCs).

  1. Importance of Vitamin D Analysis

Vitamin D may not be the primary focus of this study, but its potential influence on HNSCC biology remains important. While the biomarkers themselves did not correlate with Vitamin D levels, there was a tendency toward better overall survival (OAS) in patients with sufficient Vitamin D status. Additionally, Vitamin D has been a significant area of focus for our working group. Previous studies have demonstrated improved OAS for HNSCC patients with adequate Vitamin D levels (Bochen et al. doi:10.1080/2162402X.2018.1476817), and complex in vivo experiments in mice have shown significant immunomodulation within HNSCCs (Brust et al. doi:10.1016/j.biopha.2024.117497). These findings underscore the relevance of Vitamin D in this context. Although we did not observe statistically significant results in this study, the trends noted are valuable and contribute to a more comprehensive understanding of the role of Vitamin D in the tumor microenvironment. Moreover, presenting these non-significant findings is essential to provide a complete picture of the data and to highlight areas for future research.

  1. Scientific Integrity and Completeness

Removing 96 patients from the analysis, as suggested, would narrow the scope to a degree that compromises the integrity and completeness of our study. Our primary objective is to provide a broader view of the molecular and immunological landscape of HNSCCs, incorporating HPV+ and HPV− tumors, as well as exploring the potential influence of Vitamin D. Leaving out these aspects would not only diminish the study’s impact but also overlook results that may inspire further research, particularly on HPV− HNSCCs and their unique molecular profiles.

  1. Elucidating the Context of HNSCCs

The HPV− cohort provides critical insights into the molecular mechanisms driving HNSCCs that are not attributable to HPV-related pathways. By addressing both HPV+ and HPV− cases, our study bridges these two subtypes, emphasizing their differences and their potential implications for personalized treatment strategies. For example, the significant correlation of higher MMP-3 IRS in intratumoral regions of HPV− tumors is an important finding that aligns with the goals of understanding HNSCC molecular heterogeneity.

  1. Vitamin D as a Secondary Interest of the Study

While Vitamin D is not the primary focus, our working group has a vested interest in its role in cancer biology. Including this analysis acknowledges the ongoing exploration of Vitamin D’s influence in the tumor microenvironment and its potential relevance to immune modulation, even in the absence of direct significance.

  1. Support from Other Reviewers

The other two reviewers expressed their belief that the results presented in this study provide very interesting findings and offer meaningful insights into the molecular characteristics of HNSCCs. Their feedback highlights the value of the data and its contribution to understanding the complexity of these cancers.

In conclusion, we strongly believe that the inclusion of both HPV+ and HPV− cohorts, as well as the Vitamin D analysis, enhances the manuscript’s value by providing a more comprehensive understanding of HNSCCs. The data are not irrelevant but rather complementary to the study’s objectives, offering insights that are elucidating in the broader context of molecular oncology.

We greatly appreciate your feedback and hope this response clarifies our rationale for retaining the current structure and content. We remain open to further discussion and suggestions to improve the manuscript.

Best regards,

L. Brust

Reviewer 3 Report

Comments and Suggestions for Authors

The authors respond appropriately to the points raised by the reviewer.

Comments on the Quality of English Language

none

Author Response

The authors have addressed the reviewer’s points appropriately.